# The Relationship of Thinking Style and Motivation Features of Sales and Advertising Managers

**DOI:** 10.3390/bs10030068

**Published:** 2020-03-15

**Authors:** Alla Belousova, Yulia Mochalova

**Affiliations:** Department of Psychology, Pedagogy and Defectology Don State Technical University (DSTU), Gagarin Sq. 1, Rostov-on-don 344000, Russia; guliya@mail.ru

**Keywords:** thinking style, managers, the need for achievement, action control, motivations, interconnections

## Abstract

The thinking of a manager is conditioned by their motivational features which determine their personal professional success and the effectiveness of the organization’s activities. In this study, we assumed that two groups of sales and advertising managers had differences in the relationships between thinking style and their motivational qualities, as well as their individual need for achievement. We used the following sources: The methodology of A. Belousova for the diagnosis of thinking styles, the “scale of control over action” by J. Kuhl, and “the need for achievement” by Yu.A. Orlov. The selection consisted of 61 people, 25 to 30 years of age, of which 41 were men and 20 were women, from organizations engaged in the sale of a technical group of goods (also known as Group A) and advertising services (also known as Group B), in Rostov-on-Don. The Spearman rank correlation method was used for quantitative data processing. In group Group A, the analysis showed the presence of statistically significant connections. A critical style of thinking has a significant relationship with the level of clarity about the need for achievements and practical thinking is statistically significantly interrelated with control over action in a situation of failure. Whereas, in Group B, an initiative, managerial, and practical style of thinking has a significant correlation with the need for achievement.

## 1. Introduction

Modern production is based on using new technologies which require high level training of workers. In this context, the need for specialists that are capable of organizing and managing the production process has been increasing. Currently there is an acute shortage of professional managers, specifically, top managers who are able to understand the basics of a market-based economy and orient themselves in financial, legal, psychological issues, and especially in relationships with staff [1].

Currently, a manager is a new type of director whose main task is to efficiently organize the work of the enterprise staff. Therefore, a manager is a professional supervisor (occupying a permanent position, with people under their control) who is called upon to carry out a large amount of work that they can do only with the help of other people. A modern manager is a supervisor, a diplomat, an innovator, and a person who personally reacts to whatever is happening in their organization [2].

Managers have to be able to establish communications, coordinate the work of subordinates, delegate subordinates a range of powers, and give subordinates the freedom to defend their own opinions, while holding the control function, as well as be able to make decisions and take responsibility for their implementation. The activities of a manager include solving a diverse range of tasks, and their effectiveness is related to the thinking styles that determine the direction and originality of professional strategies chosen by them. Currently, there are specializations for the activities of managers due to the specifics of their professional tasks. The personal professional success of managers depends on how their thinking activity has developed, due to whatever motivational idiosyncrasies they have formed over time. Moreover, the effectiveness of the organization’s activities, in general, is also largely dependent on this fact [3,4].

There is a fairly extensive body of research on the professional characteristics of thinking styles. Many studies have been devoted to exploring the thinking styles of teachers [5,6,7], nurses and doctors [8,9], engineers and artists [10,11], farmers [12], students of various specialties [13,14], and representatives of ethnic groups [15,16,17].

At the same time, there is a shortage of research devoted to the study of the managerial style of thinking [18]. This area remains poorly understood, although there are studies aimed at identifying features of the style of thinking in organizations [19].

A. Belousova and V. Pishchik [20] identified three main areas in the study of thinking styles. Cognitive thinking styles include theories that emphasize the features of the image of the world, the specifics of the reflection of information, the formation of concepts, the solution of cognitive tasks, and features associated with the dominance of a certain representative system, that is, human cognitive characteristics [21,22]. Functional thinking styles are based on the use of ideas by scientists about the style of thinking as a combination of various functions [23]. Psychophysiological thinking styles refers to theories that analyze the physiological and psychophysiological determinants that define the development of a particular style of thinking [24].

A.K. Belousova’s classification of thinking styles is based on the definition of a function that each person shall assume in collaborative thinking activity [25].

According to Belousova, thinking style is one of the manifestations of systemic regulation of thinking in the psychological system. In the psychological system, first, self-organization of thinking consists of adaptive processes which are associated with the solution of existing problems and aim to preserve the system. These processes ensure the stability of the system and its self-realization within existing borders. Secondly, the processes aim to the develop the system, which goes beyond the actual boundaries of the system, as well as involving all sorts of changes due to conflicts, contradictions, and changes within the system, or the dynamics of its changes, in the process of human life. The objective level of these changes is associated with the development of the generated neoplasms. The implementation of the subject level is possible through the implementation of certain functions [25].

Thus, the style of thinking acts as a functional organization of the generation and development of neoplasms in humans and their stable and unstable manifestations. In this regard, the style of thinking can be defined as a specific set of functions, actualized by a person in various situations of problem solving.

In an adult person, thinking combines certain functions, i.e., generation, selective (selection and evaluation), sense transfer, and implementation. They are organized hierarchically and add up to a certain ratio of functions. These are fixed, constant human functions in relation to other people and to themselves, formed in the process of human development on a certain subject material. These functions are organized in people in relation to the information that comes to them during their interaction with the world.

These stable correlations of functions in individual thinking are manifested in the form of a style of thinking. Thus, in such a style of thinking, there is the resocialization, exteriorization, and individualization of the functions that have developed in man in ontogenesis, in the process of forming regulatory structures in interaction with people in the course of cognition of the surrounding world. The style of thinking is one of the regulators of thinking, which determines the dynamics and stability, and indeed the nature of the human mental activity being carried out.

In this regard, Belousova emphasizes initiative, critical, managerial, and practical thinking styles [20].

The initiative style determines the generation function, which forms the direction of thinking. People with this particular style of thinking differ in their speed of generating various ideas, are sensitive to problem situations, are unafraid to put forward hypotheses, and therefore are characterized by theoretical thinking. People whose generation function has become a personality trait are defined by qualities associated with the flexibility of switching from one idea to another, namely the “generator” who seeks to create a holistic view of the problem, and to develop a strategy to solve it.

The initiative style of thinking is characterized by a focus on the search for nonconformities, an understanding of the problem, and the setting of clear goals. This style is characterized by the initiation of thinking and the desire to understand the problem, having first located and recognized the problem.

A critical thinking style is subject to a selective function, characterized by a person’s ability to see weaknesses, mistakes, and various kinds of contradictions. Evaluative activities such as the possibility of assessment, criticism, and analysis motivate a person to these actions, since in the process, they create positive emotions. The critical thinker’s skepticism, prudence, criticality and so on emphasize of other people. Such a person easily forms new assessments, areas for criticism, and arguments. A typical characteristic associated with the critical style of thinking is the desire to assess the personality and activity of a person, their plans, and goals in general. At the same time, the results estimated are beyond those actually produced. Critical statements and evaluations dominate, which often lead to a lack of results, a product of purely intellectual activity.

For the managerial style of thinking, the function of sense transfer is characteristic. In this case, sense transfer is necessary in order to unite people around a particular case and to choose a single hypothesis, a goal, that is, a common substantive basis of the activity which then ensues. People with this style of thinking are characterized by the desire to discuss problems with others. For such people, communication is the main motivation of the activity, that is, the desire to listen to other opinions. This can be expressed in the form of organizing a discussion, as well as through the desire to work and think in joint activities with other people. In the process of discussion, people with a managerial style of thinking tend to integrate the rest of the participants, take responsibility for themselves, influence other people, and ultimately achieve the solution of the problem. An equally important quality of people with a managerial style of thinking is the ability to convey the meaning of information, reformulate it, and explain it to other participants Thus, the most distinctive feature of people with a managerial style, is their focus on others. In fact, the discovery of a new problem, the finding of contradictions, their resolution, the formulation of hypotheses and their sifting are carried out within the framework of sense transfer aimed at otherpeople.

The practical thinking style is associated with a predominance of the implementation function. A person with this style processes all the information they receive from the outside world, and primarily highlights the possibility of practical implementation. In motivational terms, they have positive emotions associated with the implementation of specific tasks, regardless of the assessments of others. People of this style are attracted to the practical implementation of ideas more than anything else. They seek to bring the problem to a logical conclusion, and only after this, can they proceed to another action. Through implementation and practice, as well as experiment and testing, they verify the value of ideas and the possibility of their real implementation. Moreover, in the course of real transformations, in practice, new assumptions and hypotheses appear and develop; their generation, however, is connected with the practical implementation of a specific idea, i.e., an idea which is thus only embodied in practice.

From the foregoing, we conclude that in the development of thinking styles, one function dominates when all the others are present. Their presence and level of development suggests that the processes of thinking and the development of neoplasms develops with any style. However, differences in thinking are, above all, differences in the dynamics and direction of the development of psychological neoplasms.

A.V. Karpov believes that finding solutions in the work of a manager is a sequential reformulation of problems and tasks. The search for solutions is also referred to as the step of generating alternatives, since it fundamentally expands the field of probable solutions, i.e., one is forced to move beyond one’s habitual thinking, so as not to miss the most effective options. To make a choice from a large number of generated solutions, one must first evaluate the solutions. A comparison of many alternatives according to many criteria, taking into account different probabilities of the implementation of solutions, is not an easy task [1] and involves the following.

The first aspect is the study of the process and strategies of multi-criteria decisions. Faced with multi-criteria comparisons, such as with a large dimension and a large amount of information, a person resorts to different heuristics which allow the comparison process at several successive stages and the processing of a certain amount of information at each stage.The second aspect, the study of human decision making, is the motivation of choice. Motivational tendency, or the intention to perform an action, is considered to be a function of the utility of the result of the action and the subjective probability (expectation) of the success of the selected action. The formation of motivation is also influenced by personal motivational determinants, i.e., achievement motivation (the ratio of the desire for success and avoidance of failure) and the locus of control (the conviction of success depending on external circumstances or on one’s own actions).

Decision making means a transition from intentions to actions, which for a manager is primarily connected with the control of actions. The motivational concept of control over action was developed by West German psychologist J. Kuhl. In this theory, there are the following four main components of the cognitive representation of intent: (1) real state, (2) required or desired state, (3) discrepancy between the required and real state, and (4) action to eliminate the discrepancy. If all components are balanced, the solution is carried out without much difficulty. Kuhl called this kind of motivation “action-oriented” [26]. In the event that during the implementation of a decision, a person distributes their attention on only some of the components or focuses it entirely on one of components, the full implementation of the action causes difficulties. This type of motivation is called “state orientation”. A “state-oriented person” analyzes the situation for a long time before making a decision, but the transition to implementation can be abandoned, and the actual implementation of the action becomes less effective and is accompanied by emotional experiences. Thus, in accordance with the ideas of J. Kuhl, for the implementation of activities, it is necessary that the subject has the appropriate motivation and ability, and also that the severity of the control of the action is taken as an integral criterion for the effectiveness of this activity. Two types of activity controls are distinguished as follows: Action orientation is a cognitive activity based on the choice of alternatives and overcoming the discrepancy between current and achieved positions; and state orientation is considered to be a cognitive activity aimed at such subjective states as those localized in the present, in the past, or in the future, presented in the form of perseveration [26].

Peculiarities of motivation constitute the most important determinant in the activity of a manager, and these same features are the most important factors in the development of the style of thinking [27,28]. A. Belousova and V. Pishchik [20] considered the system-dynamic model of motivation (M.Sh. Magomed-Eminov [29]), in accordance with which the author identified four groups of motives, i.e., motivation for initiation, motivation for selection, motivation for realization, and motivation for post-realization. Belousova and Pishchik [20] correlated these groups of motifs with the dominant functions underlying the thinking styles.

We believe that the peculiarities of the managerial style of thinking, determined by the development of certain functions, can be influenced by the motivational tendencies of a person, in particular, they can be interconnected with the peculiarities of the development of orientation towards action or orientation towards the state. The aim of this research is to assess how achievement motivation and control over action in planning, control over action in implementation, and control over action in failure correlate to different styles of thinking. Thus, the hypothesis of our study is the assumption that there is an interconnection between the managerial thinking style and motivational qualities (the aforementioned three controls over action) and the level of need for achievement.

## 2. Materials and Methods

The aim of the study was to examine, with reference to two different groups of managers, motivational qualities and the level of expression of the need to achieve. The study specifically involved organizations selling technical equipment and advertising services, in Rostov-on-Don. The survey respondents were middle managers comprised of 61 people aged 25 to 30 years, of which 41 were men and 20 were women. The participants of the empirical research were managers of different fields of activity, namely, managers who directly carried sales of technical equipment, and managers of organizations engaged in marketing and advertising.

Needless to say, advertising managers and sales managers are distinguished by the nature of their professional activities [30]. An advertising manager plays a key role in, among other things, promoting various business offers and forming a loyal target audience. The manager develops strategies and creates products that will clearly attract the attention of potential buyers and will stimulate the interest of new customers. The main activity of the sales manager, meanwhile, is to interact directly with customers, and as such an employee they are the last link in the chain of operations leading to a purchase [30]. Differences in professional activity determine differences in cognitive activity, in particular thinking styles, and in the motivational features of advertising managers and sales managers [31,32,33,34,35].

The following employees of several organizations took part in this research:Group A was comprised of a group of managers engaged in the sale of technical equipment. Two organizations engaged in the sale of heat, water, and gas equipment. There were 30 people (22 men and 8 women). Middle managers engaged in sales, both in the sales area and directly in the office, which involved working with the client on the phone.Group B was comprised of managers of organizations engaged in advertising and marketing activities and including two organizations engaged in advertising activities. There were 31 people (19 men and 12 women).

As the hypothesis of the study, we assumed that there is an interconnection between the managerial thinking style and motivational qualities (control over action in planning, control over action in implementation, control over action in failure) and the level of need for achievement. In the course of the study the following methods were used: theoretical and empirical analysis, generalization, systematization, schematization, conversation, testing, and mathematical methods of data processing. The following research methods were used: The questionnaire “Styles of thinking” (Belousova) diagnosed the styles of thinking as initiative, critical, managerial, and practical. It consisted of 32 questions, forming four scales of 8 questions, corresponding to four styles of thinking. Each question implied differentiated answers, i.e., “yes”, “rather yes”, “rather no”, and “no”. The respondents chose the answer in accordance with their opinion [20].The questionnaire by Kuhl’s “Scale of control over the action” (Russian version) included the scales “control over the action during planning”, “control over the action during the implementation”, and “control over the action during the failure.” The method contained 36 situations. Each situation had two response alternatives, one alternative related to action-oriented behavior and the other to state-oriented behavior. To calculate test scores for each subject, it was necessary to summarize the answers corresponding to each scale. Kuhl identified four strategies for controlling actions as follows:
Attention control “action-oriented” is where one is fully focused on the task and does not pay attention to irrelevant aspects of the situation;Motivational control, i.e., controlling one’s own motivation is where an “action-oriented” manager retains interest only in information that supports the decision and does not provoke doubt.Emotional control “orientation toward the state” means a greater focus on experiences and the corresponding blocking of actions.Failure control, after failure to implement solutions, “state-oriented” solutions are ineffective with new attempts to solve, because they focus on experiencing the past situation, rather than on the actual one, which leads to short-term memory overload and errors [36].The questionnaire “The need to achieve the goal” (Orlov) contained 23 questions to determine the significance of the attitude toward achievement as a whole, and to diagnose the level of the individual’s need to achieve the goal [37].

The reliability of the results was ensured by a sufficient sample size, along with validity and reliability of the methods used. To achieve representativeness of the sample, we used a simple random strategy to select subjects, which consisted of randomly extracting objects from the general population of objects to create equivalent groups [38]. In order to study the relationship of these parameters, we used the Spearman’s rank correlation method (Table 1), a correlation study being conducted using the SPSS statistical package (IBM SPS Statistics Version 20) [38].

## 3. Results and Discussion

Let us first analyze the features of the relationship of thinking style, the level of severity of the need to achieve, and control over the action in Group A.

Analyzing the data given in the table, we note the presence of statistically significant relationships. Thus, the critical thinking style has a significant relationship with the level of expression of the need for achievement (r = 0.468, r_cr_ = 0.47, and *p* = 0.01). Consequently, as the need for achievement in this group of respondents increases, the dominance of the critical style of thinking increases along with it. Practical thinking is statistically significantly interrelated with control over the action in a situation of failure (r = 0.417, r_cr_ = 0.36, and *p* = 0.05). We assume that by increasing attention to results of the work done, respondents will be more successful, proceeding without mistakes and defects to realize further new ideas. It should be noted that the control over the action during the implementation is negatively related to all the thinking styles of respondents in this group, but also that this relationship is not statistically significant. This can be explained by the excessively stereotyped actions of managers which do not contribute to the development of thinking.

Thus, in the group of sales managers with increasing control over the action in case of failure, the implementation function becomes more significant. The significance of selective function increases with the level of need for achievement.

Let us consider the features of the relationship of thinking style, the level of expression of the need for achievement, and control over the action in Group B. In order to study the relationship of these parameters, we used the Spearman’s rank correlation method (Table 2).

Analyzing the data given in the table, we note the presence of statistically significant relationships. Thus, the initiative style of thinking has a significant relationship with the level of expression of the need for achievement (r = 0.384, r_cr_ = 0.36, and *p* = 0.05). Consequently, as the need for achievement in this group of respondents increases, the dominance of the idea generation function increases simultaneously. The level of expression of the need for achievement is significantly connected with the managerial and practical thinking styles (r = 0.402, r_cr_ = 0.36, *p* = 0.05 and r = 0.581, r_cr_ = 0.46, *p* = 0.01, respectively). For managers in the field of advertising and marketing activities, therefore, it is typical with the growing need for achievement, to increase the desire to realize the idea and do it collectively.

Practical thinking style is statistically significantly interrelated with control over action in a situation of failure (r = −0.412, r_cr_ = 0.36, and *p* = 0.05). We assume that increased attention to the results of the work done does not contribute to the implementation of specific tasks, regardless of the assessments of others. It should be noted that the control over the action during the implementation is negatively related to all the thinking styles of the respondents of this group, and only the connection with the managerial thinking style is at a statistically significant level (r = −0.438, r_cr_ = 0.36, and *p* = 0.05). Consequently, the increase in the prevalence of management style reduces the control over the action during the implementation. It should also be stressed that the focus of thinking of the respondents is on a single process, either the collective solution of the problem or the monitoring of the process operation. The results can be explained by the specifics of the professional activity of managers in the field of advertising, which requires non-standard solutions and actions.

## 4. Conclusions

In Group A, the analysis showed the presence of statistically significant connections. Critical thinking style has a significant relationship with the level of expression of the need for achievement. Meanwhile, the practical thinking style is statistically significantly interrelated with the control over the action in a situation of failure. Thus, in the group of sales managers with increasing control over the action in case of a failure, the implementation function becomes more important, and practical thinking along with it. The importance of selective function increases with the level of need for achievement, while critical thinking style increases in the same circumstances.In Group B, an initiative, managerial, and practical thinking style has a significant correlation with the need for achievement. The practical thinking style is statistically significantly interrelated with control over the action in a situation of a failure (negative correlation). The managerial thinking style has a negative relationship with control over the action during the implementation. Therefore, increasing the predominance of the management style reduces the control over the action during the implementation. Noteworthy is the concentration of respondents‘ thinking on a single process, the collective solution of a task, or tracking the process of performing an action. The results obtained can be explained by the specifics of the professional activity of managers in the field of advertising, which requires non-standard solutions and actions.

Thus, in the group of advertising managers, when the need for achievement increases, opportunities for an initiative, managerial and practical style of thinking expand. In a situation of failure, increased control over the action reduces the practical style of thinking; in a situation of implementation, with increased control over the action, the style of managerial thinking decreases.

The obtained results confirm our hypothesis that the style of thinking has a close relationship with the need for achievement. G. Murray (1938) was one of the first to show the importance of achievement motivation, highlighting two components, i.e., the need to achieve success and the need to avoid failure. Following this, the motivation for achievement was studied by D. Atkinson (1964), D. S. McClelland (1988), and H. Heckhausen (2008) [39]. Modern research highlights the relationship between thinking styles and achievement motivation in Chinese students [27] and in Iranian students [28]. Our research shows similar results, given that both groups of managers found direct correlations of the need for achievement with the initiative, management, critical and practical style of thinking. However, whereas, in Group A, significant relationships exist with the critical thinking style, the other types of thinking styles have no significant correlations with the need for achievements. At the same time, in the Group B of managers, while there are significant direct correlations of the need for achievements with the initiative, management and practical style of thinking, there are no significant relationships with the critical style of thinking. Thus, it is argued that for managers of different professional orientations, the relationship between the style of thinking and the need for achievements is characteristic. This confirms that the need for achievement and achievement motivation is one of the determinants of managers‘ thinking style.

According to J. Kuhl’s motivational concept of action regulation [26], the key elements of intention are the processes of initiating an action, representing the goal and methods of achieving it, and analyzing past and future events that could affect the implementation of the action. The representation and activation of these components form the basis of a full-fledged intention, which ensures the successful execution of the action. Full intent involves control in three areas, i.e., control over the action during the planning, control over the action during the implementation, and control over the action during the failure.

When we talk about the style of thinking, we mean the implementation of thought actions, therefore, the question of the specific types of motivational control implemented is significant for us.

In Group B, managers engaged in advertising and marketing activities have inversely proportional relationships of practical thinking style and control in the case of failure, which means that the more difficulties and failures they encounter, the less managers try to implement mental tasks. The presence of a negative correlation between the management style of thinking and the control over the action during implementation, means that the more their managerial style of thinking develops, the less they are able to keep the current intention in focus, and to show perseverance.

The study confirmed the importance of action orientation in the structure of motivational regulation. In Group A, sales managers showed significant correlations between a practical style of thinking and control in the case of failure. Failure control reflects the ability of managers to initiate the process of implementing an intention, despite the difficulties that accompany it. This means that the practical style of thinking involves managers performing mental tasks regardless of failures and difficulties.

## Figures and Tables

**Table 1 behavsci-10-00068-t001:** Table of correlation relations of thinking style and motivational tendencies in Group A.

No. i/s	Thinking Styles	Need for Achievement	Control over the Action during the Failure	Control over the Action during Planning	Control over the Action during the Implementation
1.	Initiative (I)	0.153	0.132	0.089	−0.098
2.	Critical (C)	0.468 *	0.147	0.285	−0.061
3.	Managerial (M)	0.278	0.082	−0.022	−0.035
4.	Practical (P)	0.154	0.417 **	0.308	−0.129

Note: *, correlation coefficient is significant at *p* = 0.01; **, correlation coefficient is significant at *p* = 0.05.

**Table 2 behavsci-10-00068-t002:** Table of correlation relations of thinking style and motivational tendencies in Group B.

No. i/s	Thinking Styles	Need for Achievement	Control over the Action during the Failure	Control over the Action during Planning	Control over the Action during the Implementation
1.	Initiative (I)	0.384 **	0.019	0.156	−0.075
2.	Critical (C)	0.351	−0.151	−0.239	−0.294
3.	Managerial (M)	0.402 *	−0.174	−0.301	−0.438 **
4.	Practical (P)	0.581 *	−0.412 *	−0.176	−0.196

Note: *, correlation coefficient is significant at *p* = 0.01 and **, correlation coefficient is significant at *p* = 0.05.

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
