# Peer review of "The Relationship of Thinking Style and Motivation Features of Sales and Advertising Managers"

_behavsci, 2020, doi:10.3390/bs10030068_

Round 1
Reviewer 1 Report
The idea of research are very interesting and meaningful not only for scientists but also for practitioners, but the quality of the manuscript is low and needs rewriting.
The authors write (begins in 46 line) "Manager's personal professional success depend on the fact how his thinking activity is developed, due to his formed motivational peculiarities. In general, the effectiveness of the organization's activities is also largely depending on this fact", but there are no justification, references.
In general, the Introduction focuses very much on describing thinking styles, but poorly explains what motivation is. There are no explanation about different tendencies of motivation (failure control, planning control, etc.). It is not clear why thinking styles are associated with motivation. The problem is that the idea of the research is not clear: do authors want to evaluate which thinking style is dominating for concrete group of managers, or how achievement motivation correlates with different styles of thinking or...?
It is the problem with use of definitions, concepts in the text. For example, the authors write (line 92) "A. Belousova emphasizes proactive, critical, managerial and practical thinking styles...", then in the following text use definition The initiative style. There are no explanation if proactive and initiative styles are the same or different.
Hypothesis is written differently in lines 181 and 203.
The research methods are not described.
The authors use words Experimental base (188 line), but they are not suitable for correlational research.
There are no explanation why the results are presented separately for groups A and B, how they should differ.
Some sentences are repeated in the text, for example, see lines 215 and 223.
Author Response
Пожалуйста, смотрите вложение

Reviewer 2 Report
The study is well-done and the manuscript is well-written. Two items should be reviewed:
209-210 "The reliability of the results was ensured by sufficient sample size,
validity and reliability of the methods used." -- no support is supplied for this claim. The document will be stronger if you cite the support for the sample size and Spearman analysis conducted.
234 - 'steretyped' - check spelling, please
Author Response
Please, see the attachment

Round 2
Reviewer 1 Report
Lines 207-208: "differences in professional activity determine differences in cognitive activity" - are you sure??? Who proved that???!!!
Author Response
Please see the attachment

This manuscript is a resubmission of an earlier submission. The following is a list of the peer review reports and author responses from that submission.